# Development of Fibre-Reinforced Cementitious Mortar with Mineral Wool and Coconut Fibre

**DOI:** 10.3390/ma15134520

**Published:** 2022-06-27

**Authors:** Paul O. Awoyera, Oluwaseun L. Odutuga, John Uduak Effiong, Astelio De Jesus Silvera Sarmiento, Seyed Javad Mortazavi, Jong Wan Hu

**Affiliations:** 1Department of Civil Engineering, Covenant University, Ota 112233, Nigeria; paul.awoyera@covenantuniversity.edu.ng (P.O.A.); seunodotuga@gmail.com (O.L.O.); effiong.john@yahoo.com (J.U.E.); 2Fundacion Para La Investigacion, Desarrollo E Innovacion, Barranquilla 586790, Colombia; asilvera@fundacionidi.org or; 3Universidad de la Costa, Barranquilla 585566, Colombia; 4Department of Civil and Environmental Engineering, Incheon National University, Incheon 22012, Korea; mortazavi@inu.ac.kr; 5Incheon Disaster Prevention Research Center, Incheon National University, Incheon 22012, Korea

**Keywords:** coconut fibres, mineral wool fibres, mortar, durability, strength properties

## Abstract

Globally, as human population and industries grow, so does the creation of agricultural, industrial, and demolition waste. When these wastes are not properly recycled, reused, or disposed of, they pose a threat to the environment. The importance of this study lies in the beneficial use of coconut fibre and mineral wool in the form of fibres in cement mortar production. This study examines the use of coconut and mineral wool fibres in the production of fibre-reinforced mortar. Five different mortar mixtures were prepared, having one control mortar along with four fibre-reinforced mortars. The control mortar is denoted as CM while 1% and 1.5% of mineral wool are incorporated into this mortar mix and denoted as RMM-1.0 and RMM-1.5, respectively. Additionally, the mortar sample configurations contain 1% and 1.5% coconut fibers, designated as RCM-1.0 and RCM-1.5. These samples were subjected to different strength and durability tests to determine their suitability for use in mortar production. The testing findings show that mortar containing 1.5% mineral wool has better compared flexural strength and durability properties. The investigation results will form part of the database for the efficient utilization of natural and waste fibres in the construction and building sectors.

## 1. Introduction

The generation of waste from farmlands, industrial, and building demolition activities is gradually increasing. These waste materials pack landfills and are sometimes found in water bodies. This constitutes a major environmental menace. Natural fibres are some of the waste materials that lead to severe contamination problems. Dropping such waste in open land areas endangers the environment by contaminating water and air [1]. The first waste products generated during the fabric manufacturing process are fibres that do not pass through the sorting phase. These fibre wastes, according to [2] account for 15% of the overall waste generated throughout the production process. Some of these natural fibres include coconut fibres [3,4,5], mineral wool fibres [6], bagasse fibres [7,8], and kenaf fibres [9]. Indonesia is the world’s largest producer of coconuts, accounting for 31.2% of global output. Both the Philippines (25.5%) and India (17%) are in the top ten. Coconut is also dominant in Nigeria and some other sub-Saharan African countries. Coconut production reached 62.14 million metric tons in 2012, according to estimates from throughout the world. Coir produced annually by this number of coconuts could exceed 12.2 million tons [5]. Mineral wool waste is merely a minor fraction of the overall building demolition debris in terms of mass. In contrast to other types of construction and demolition waste, the low bulk density of this material requires considerably less transportation and landfilling capacity. By using this material, not only will landfilling issues be alleviated, but virgin raw resources will be saved [6,10]. This incentivizes the adoption of proper measures to tackle the menace created by agricultural waste.

There is a growing usage of cementitious composite materials such as mortar for various construction work, such as flooring, wall finishes, structural repairs, and more. The growing demand for the use of mortar has given rise to the need for various forms of improvement of cementitious composite materials to provide better efficiency in construction. When mortar is subjected to various forms of loading, micro-cracks begin to appear along planes that experience tensile strains. Further application of loads causes cracks to grow uncontrollably. The inclusion of fibres can be used to mitigate the growth of these cracks [11]. In other words, the utilization of fibres in mortar mix is seen as one of the major improvements in mortar production and utilization. Fibres have been proven to be quite effective in improving the flexural and split-tensile capacity of cementitious composite materials. Fibres ought not to be viewed as a substitution for traditional reinforcement bars, despite the fact that in certain applications, this might be the situation. They are reciprocal strategies for reinforcing mortar, and there are numerous applications where they should be utilized together [12]. However, the majority of the fibres utilized in the present day are synthetic in nature, and possess high embodied energy with low sustainability prospects. Most of these synthetic fibres present ecological [13,14] and medical issues [15] during their production and usage. Furthermore, the cost of these human-made fibres (glass fibres, polypropylene fibres, and steel fibres) is high [16,17]. This has given rise to the need for innovative, sustainable, and natural materials that can be incorporated in mortar to improve the mechanical and structural properties, while taking into account the need for greener construction. The feasible and sustainable improvement of buildings will require the incorporation of natural and ingenious materials, and the reusing of waste materials, which constitute a large percentage of landfill mass and have ecological consequences [18]. At the moment, research has been led on the utilization of common plant fibres (banana, sisal, hemp, flax, jute, coconut, and oil palm) in mortar. Most of this research has portrayed natural fibres as prospectively being able to enhance the mechanical and strength properties of mortar. Studies have suggested that these fibres can help delay or stop the propagation of micro-cracks in cementitious composite materials, improve the post cracking behavior of mortar, expand the protection from dynamic loads, improve ductility and toughness, and lower the perviousness of mortar, thus reducing the loss of water [19,20,21]. The reuse of natural fibres in the construction industry would safeguard natural assets that cannot be replenished, diminish the ecological contamination caused by solid wastes, and preserve energy that would otherwise have been utilized during incineration processes [22]. For effective utilization of these fibres, studies have also shown that treated natural fibres perform better in enhancing the strength and mechanical properties of cementitious composite materials when compared to untreated ones. These fibres can be treated using natural solvents, water, and dilute alkali [23]. Improved surface morphology and decreased non-cellulosic fiber content are two additional advantages of this chemical treatment. Surface treatment, for example, improves the structure of natural fibres. It has been shown that alkaline treatment (NaOH) increases fibre surface roughness by disrupting hydrogen bonding. Short-length crystals are exposed and the cellulose component is depolymerized by the alkaline treatment, which removes the oils, waxes, inorganic salts, and lignin that coat the fibre’s outer surface. The strength of composites can also be affected by excessive fibre concertation treatment, which can cause fibre surface rupture, damaging the primary and secondary walls of fibre [17]. Utilization of these fibres instead of engineered synthetic fibres in mortar or other materials has special benefits, such as lessening carbon footprints and helping build viable solid waste administration systems.

Agricultural wastes have been continually utilized in cementitious composites. However, the focus of this study is to explore the synergistic effects of both coconut fibre and mineral wool on the mechanical properties of mortar. This study is novel, as the use of these materials in this context has not been seen in any prior literature, to the best of our knowledge. Additionally, these materials are readily available in the region in which the research is conducted. Moreover, the study aims to promote sustainable mortar production for buildings. Considerable importance is placed on studying the physical, mechanical and microstructural properties of these fibres, in order to determine the feasibility of their use as eco-friendly alternatives to currently used cement-mortar reinforcement fibres. This could potentially lead to advances in the search for economical building-structure reinforcement materials.

## 2. Materials and Methods

Ordinary Portland Cement of grade 42.5 was utilized as the binder material in the preparation of the mortar specimens. Natural fine sand was adopted for the mixture in combination with cement, with a mix proportion of 1:3, and a water–cement weight ratio of 0.55 The natural fibres (coconut and mineral wool fibres) utilized in making the cement mortar samples were obtained from Ota, Ogun State, Nigeria. The properties of these natural fibres are shown in Table 1. Stone wool is the specific type of mineral wool used. This is evident in the physical appearance and the element compositions as seen on the EDS analysis, with the increase in key element compositions such as P, Mn, and Fe. These elements help draw differences between stone wool and glass wool, aside from their physical properties. Furthermore, the inference of mineral wool being natural in this study has been corrected to show its synthetic origin, due to its manufacturing process.

Sieve analysis was conducted in line with BS EN 1015 Part 1 [24]. The gradation of the fine aggregates of the sieve analysis is shown on Figure 1.

Fibres were alkaline treated after they were dried and straightened in order to enhance their mechanical qualities. A 6% NaOH solution was used to soak the fibres for 2.5 h at room temperature, and the fibre to solution ratio was 50 g:300 mL. For 48 h, the fibres were dried in an oven set to 80 °C after being properly rinsed until they reached a neutral pH value. After treatment, fibres were then distributed and mixed equally throughout the mixture, similar to fibre-reinforced concrete [25], in line with the sample configurations on Table 1. Proper mixing was achieved by adding the fibres before adding the water, as shown in Figure 2. After casting and allowing the samples to set for 24 h, the samples were demoulded and allowed to cure in water for 7, 14, and 28 days. The fresh and hardened properties of the mortar samples were measured according to BS EN 1015 Part 1, 6, 11, 18; BS 812 Part 2; and BS EN 12350 Part 2 [24,26,27,28,29,30]. These tests include: sieve analysis on fine aggregates, specific gravity and compacted bulk density of fine aggregates (prior to mixing), slump test on fresh mortar, flexural and compressive strength test, and bulk density and water absorption tests on hardened mortar. Furthermore, microstructural tests were conducted on the selected hardened mortar samples to study the morphology and determine the mineral composition of the structure of the composite material.

Fifteen different mortar samples per test (compressive strength, water absorption, bulk density) were prepared, which included three control mortar samples and twelve fibre-reinforced mortar samples, which had 1.0% and 1.5% of mineral wool and coconut fibres incorporated, in line with the desired mix of configurations, as shown in Table 2. For the flexural test samples shown in Figure 3, ten mortar samples were used per test, with dimensions of 40 × 40 × 160 mm. 

## 3. Results and Discussion

### 3.1. Test on Fresh Mortar

#### Slump Test

Slump, which is a critical physical quality that must be maintained in mortar [31], was tested on samples to evaluate the workability of the mortar. From the slump test results, as shown in Table 3, the control mix had the highest slump value of 50 mm. An addition of 1% of mineral wool fibres and coconut fibres resulted in a reduction of slump value by 20%, while the addition of 1.5% of mineral wool fibres and 1.5% of coconut fibres achieved a reduction in workability by 40% and 30%, respectively. The reduction in workability of the mortar is due to the higher water absorption characteristics of these fibres, as also observed from previous studies [32,33]. Generally, the workability of cementitious mixtures depends largely on the water absorption capacity of raw materials, and also on the cohesion of particles. In this context, mineral wool fibre exhibits higher water absorption (about 8%) than coconut fibre (about 5%), and becomes soft, thereby reducing the cohesion of the matrix in the wet state. Moreover, it is noteworthy that water absorption increases with an increase in fibre content, thus reducing the workability of the mixture. This was the case when fibre content was increased from 1 to 1.5%.

The workability of mortar is not widely reported in the open literature, as tests are only performed according to the proposed applications of a mortar. In this study, the exhibited workability of tested mortars is within an acceptable limit for building mortar.

### 3.2. Test on Hardened Mortar

#### 3.2.1. Flexural Test on Mortar

It was observed that RMM-1.5 achieved the best flexural strength with a 47% increase at 28 days in comparison with the control mix. It was observed that all the fibre-reinforced mortar mixes at 28 days attained more than 75% of the control mix flexural strength, which is impressive. RCM-1.0 achieved a 23% increase at 28 days in comparison with the control mix, making it the second-best mix in the flexural strength test series. The flexural test results are represented in Figure 4.

#### 3.2.2. Compressive Strength Test

In comparison to the mineral wool and coconut fibre-reinforced mortar samples, the control mortar had higher compressive strength than the modified mortars. It was observed that RMM-1.0 achieved the best compressive strength with a 12% decrease at 28 days in comparison with the control mix. It was observed that RCM-1.5 attained just 46% of the control mortar mix compressive strength. Hence, RCM-1.5 should not be considered for structural work. It was also observed that RMM-1.5 finished as the second-best mix in this compressive strength series with a 32% decrease at 28 days in comparison with the control mix. The decrease in compressive strength could be due to fibre overlapping and creating voids, causing the pathway to fail during testing. The higher the fibre content, the lower the compressive strength. This is also observed in previous studies that utilized plant fibres [22,32,33,34]. The compressive strength test results are represented in Figure 5. As expected, different fibres caused variations in mechanical performance. This is because the fibres exhibit varying mechanical characteristics, which, in turn, influenced the overall mechanical properties of the mortars.

#### 3.2.3. Water Absorption Test

This test was carried out by removing the saturated mortar cube from the curing water tank, then measuring and recording the weight of the saturated mortar sample. The saturated mortar samples were oven-dried for at least 24 h. A measurement of the weight of the dry mortar sample was recorded. The water absorption was then calculated using the formula:(1)$$\frac{\mathrm{Wet}\;\mathrm{weight}-\mathrm{dry}\;\mathrm{weight}\;}{\mathrm{Dry}\;\mathrm{weight}\;}\;\times \;100$$

The water absorption test from Table 3 shows that RMM 1.0 at 7 days shows the least water absorption capacity, while RM 1.5 shows the greatest water absorption capacity. The water absorption test for samples after days of curing shows RM-1.5 with the least water absorption capacity, while the control mix exhibits the greatest water absorption capacity. The water absorption test results are shown in Table 4 and Figure 6.

#### 3.2.4. Bulk Density Test

The bulk density of concrete reflects concrete compaction and the potential of concrete to work effectively for structural strength, water and solute movement, and durability [35]. The bulk density of various specimens is shown in Figure 7. It is observed from the experimental results that the control specimen had a bulk density of 18.6 Kg/m^3^, higher than the rest of the specimens, as shown in Figure 7. It is also observed that, as the weight content of fibres increases, the bulk density of the concrete decreases. This gradual decline is somewhat similar to the decline in compressive strength, showing how the compactness of the concrete may affect its compressive strength. Changes in the bulk density for the samples ranged from 3.8% to 6.5% compared to the control.

#### 3.2.5. Microstructural Analysis

For the microstructural analysis, the control concrete and the best mix at 7 days and 28 days of curing were examined. RMM 1.5 was chosen as the best mix, as it achieved the best results in the flexural strength test and the second-best result in the compressive strength test. Further conclusions can be drawn to validate the results based on the morphology of the samples, as shown in Figure 8 and Figure 9. From the Scanning Electron Microscopy chart, the control sample showed a quite compact but rough surface, with fewer pores compared to the more refined surface of the RMM1.5 sample. The RMM1.5 sample, however, had larger micro-cracks compared to the control. These physical properties may have contributed to the higher compressive strength and lower water sorptivity observed with the control sample when compared to RMM1.5 at 7 days of curing. However, at 28 days of curing, RMM1.5 showed an all-around refined and smaller cloudy surface, though with a couple of micropores, when compared to the control sample, which showed a rough surface with a bit of material segregation. This difference in physical properties may also have contributed to the higher flexural strength and lower water sorptivity of the RMM1.5 sample when compared to the control sample. In other words, it is clear that the hydration cycle of the blended 1.5% of mineral wool in mortar was adequate, which additionally upheld the upgraded strength properties seen in the mixes.

As shown in Figure 10, calcium and silicon were the most dominant compounds in the control mix. The presence of these components is fundamental in the utilization of aggregates for their non-reactivity in ordinary situations and for their toughness, which makes them appropriate for expanding substantial strength and sturdiness. It can be said that the strength gained via the addition of fibres can be influenced by the high levels of these elements. Other elements present in the structure include aluminum, manganese, and iron, mostly in the modified mixtures.

## 4. Comparison of Results with Previous Studies

Previous studies have been conducted in ascertaining the performances of natural fibre cement mortars, as shown in Table 5 and Table 6. These comparisons were done based on parameters such as the compressive strength and flexural strength of cement mortars utilizing natural fibres.

In terms of compressive strength, most studies recorded a decline in compressive strength values with the increase in natural fibre content, as shown in Figure 11. The present study shows a decline in 7-day compressive strength, ranging from 13.29–46.6% with the addition of mineral wool (1–1.5%). An increase in the addition of coconut fibres (1–1.5%) showed the greatest reduction in compressive strength (46.68–66.77%). This behaviour of natural fibre cement mortar with respect to compressive strength is slightly improved after 28 days of curing, though still of lesser compressive strength than the control specimen, as seen in Table 4. The decrease in compressive strength ranges from 12.0–31.99% with an increase in mineral wool content (1–1.5%) while the addition of coconut fibres (1–1.5%) led to a reduction in compressive strength of the mortar (40–56%). However, a study by Mathavan et al. [32] on using organic fibres of wool and silk shows a decline of 4.55% in compressive strength from the control at 7 days with 1.0% fibre content, but an increase in compressive strength by 2.84% with 1.5% of organic fibres at 7 days. The improvement in compressive strength is even greater after 28 days of curing, as seen in Table 4 [32]. In the study by Chandrasekaran et al. [33], coconut fibre cement mortar with a control grade of 58 MPa was also reduced in compressive strength when compared to normal mortar. The decline ranged from about 35%–41% for samples cured for 28 days with a fibre content of 1% to 1.5%, respectively. The reduction in compressive strength of sisal-fibre cement mortar is slightly more (43–48%). Palmyra-fibre cement falls almost within that range of strength decline in compression (41–50%). Banana-fibre cement with a similar fibre content declined in compression capacity by 45–55%. Spikelet oil palm fibre shows a smaller reduction in compressive strength values (2.73–14.29%) according to studies conducted by Rama Rao & Ramakrishna [34]. Studies carried out by Piña Ramírez et al. [22] show that, with addition of 30–50% of mixed waste of mineral wool, the decline in compressive strength is managed between 11% and 13.2% with cement mortar of grade 25/30. An in-depth analysis into the physical, mechanical, and structural properties of these fibres might be key to ascertaining the differences in performances, especially where mortar grades and fibre content are quite similar.

In terms of flexural strength, most studies recorded an improvement in flexural strength values with an increase in natural fibre content, as shown in Figure 12. The present study shows a decline in 7-day flexural strength with the addition of 1% of mineral wool by 7.73%, while the addition of the same fibre content using coconut fibres yields an increase of 44.3%. However, when the fibre content for mineral wool was increased to 1.5%, the 7-day flexural strength was improved by 5.8%, while a decline of −47.3% was observed with the addition of 1.5% of coconut fibres in cement mortar. The flexural strength of mineral-wool-fibre cement mortar with 1% fibre content was reduced by 5.9% after 28 days of curing, while a similar content of coconut fibres improved the flexural strength by 23.7%. At 1.5% of fibre volume, mineral wool improved the flexural capacity of the mortar by 47.1%, while the addition of the same fibre volume of coconut fibres yielded a decline in flexural strength of 23.6% after 28 days of curing. However, a study by Mathavan et al. on using cotton and linen plant fibres shows a decline in flexible strength ranging from 8% to 14.6% with the addition of 0.5% and 1% of the fibres compared to the control at 28 days. With 1.5% plant fibre content, flexural strength was enhanced by 18.8%. The flexural performance for the organic fibres was such that a fibre content of 0.5% to 1.0% yielded a decline in flexural capacity (8.4% to 1.9%). This was inverse to the case where, with an increase of fibre content up to 1.5%, the flexural strength increased by 10.23%. In the study by Chandrasekaran et al. [33], coconut-fibre- (1–1.5%) cement mortar improved flexural strength by 8.1% to 32.4% in comparison to normal mortar, contrary to the present study. The increase in flexural strength of sisal-fibre-cement mortar is slightly more (27–46%). Palmyra-fibre-cement mortar falls almost within that range of flexural capacity strength increase (16–46%). Banana fibre cement with a similar fibre content also increased in flexural capacity by 27 to 32.4%. Spikelet-oil-palm fibre also shows an increase in flexural strength values (12.2–61%), according to studies conducted by Rama Rao & Ramakrishna [34], with an increase in fibre content within 1% to 2%. Studies carried out by Piña Ramírez et al. [22] show that, with the addition of 30–50% of mixed waste of mineral wool, there is a slight improvement in the flexural capacity by 2.13%, but this changes with an increase in fibre content up to 50%, as a decline between 3.2% and 1.4% is seen with cement mortar of grade 25/30. An in-depth analysis into the physical, mechanical, and structural properties of these fibres might be key to ascertaining the differences in performances, especially where mortar grades and fibre content are quite similar.

In terms of water sorptivity, the present study shows that an increase in mineral wool and coconut fibres causes a decrease in water-sorptivity capacity, which is a positive attribute, improving the durability of the mortar. The present study shows a decline in 28-day water sorptivity with the addition of 1% of mineral wool by 10.11%, while the addition of the same fibre content using coconut fibres yielded a decrease of 43.82%. However, when the fibre content of mineral wool was increased to 1.5%, the 28-day water sorptivity was further reduced by 82.6%, while a decline of 64% was observed with the addition of 1.5% of coconut fibres in cement mortar. However, a study by Mathavan et al. on the use of cotton and linen plant fibres shows a 38% to 147% increase with the addition of 0.5–1.5% fibres compared to the control at 28 days. This attribute connotes the unsuitability of this sort of mortar in terms of durability in prohibiting the ingress of water.

## 5. Conclusions

This study focused on the feasibility of using coconut fibre and mineral wool for the production of fibre-reinforced mortar. The mechanical properties of the mortar samples prepared with the incorporation of the fibres (mineral wool and coconut fibres) were compared to the mechanical properties of conventional mortar. Specifically, the compressive strength, split-tensile strength, and water absorption of all the prepared mortar samples were tested and compared to determine the effects of the inclusion of the natural fibres as mortar production materials. The following conclusions were drawn from this study:Workability of the fibre-reinforced mortars was lower than that of the control mixtures, which was attributed to the varying water absorption capacities of the added fibres, thus making the matrix less cohesive at the wet state.Overall, the mechanical test results showed that excessive fibre content or the fibre orientation in the matrix contributed largely to the strength properties of the mixture. In this study, the best strength performance for the mortars was achieved at a fibre content of 1%. The study also deduced that the overlapping of fibres (in case of higher fiber volume) could create failure pathways in the material under test load.The compressive strength and flexural strength of mortars produced in this study were best at a fibre content of 1.0%.This study recommends RMM-1.5 to be implemented as a viable mortar mix design as it is also a more optimal option in terms of strength and durability.Additionally, further research is crucial to decipher possible enhancements to the reduced compressive strength of natural-fibre-reinforced concrete.To further improve the performance of mortars using fibres, research in the durability assessment of mortars incorporating natural fibres is encouraged.

## Figures and Tables

**Figure 1 materials-15-04520-f001:**
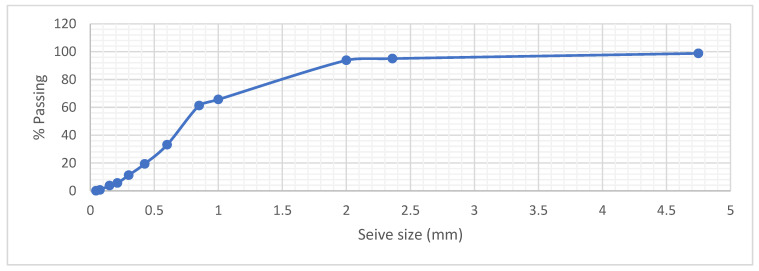
Gradation Distribution of utilized fine aggregates (with fineness modulus of 6.12; specific gravity of 2.578 and water absorption capacity of 1.13%).

**Figure 2 materials-15-04520-f002:**
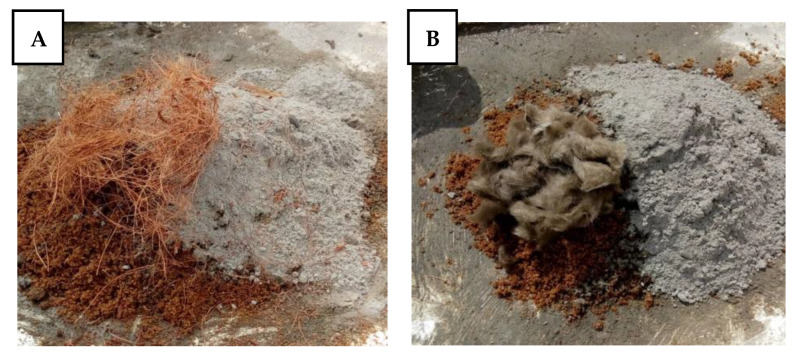
Mix configurations: (**A**) Coconut fibres + fine aggregate + cement; (**B**) Mineral wool fibres + fine aggregate + cement.

**Figure 3 materials-15-04520-f003:**
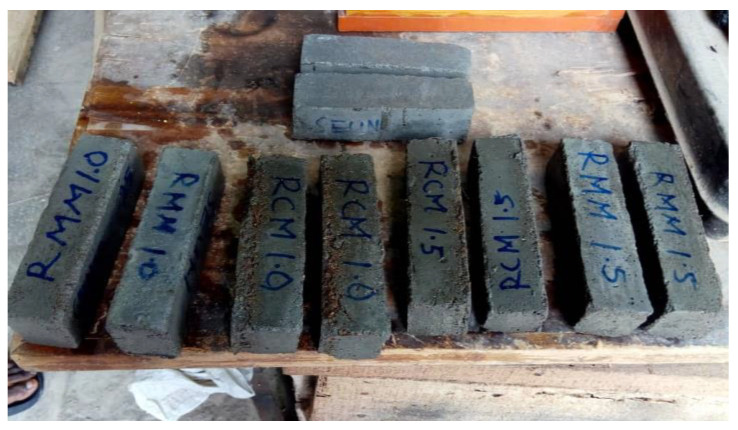
Hardened mortar samples.

**Figure 4 materials-15-04520-f004:**
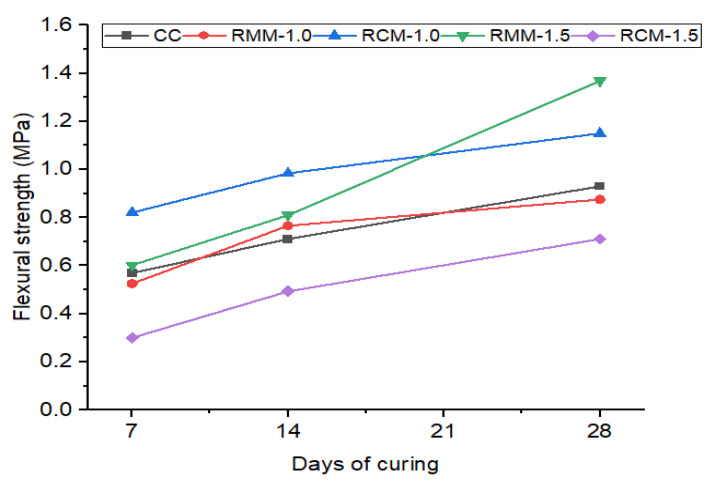
Flexural test results.

**Figure 5 materials-15-04520-f005:**
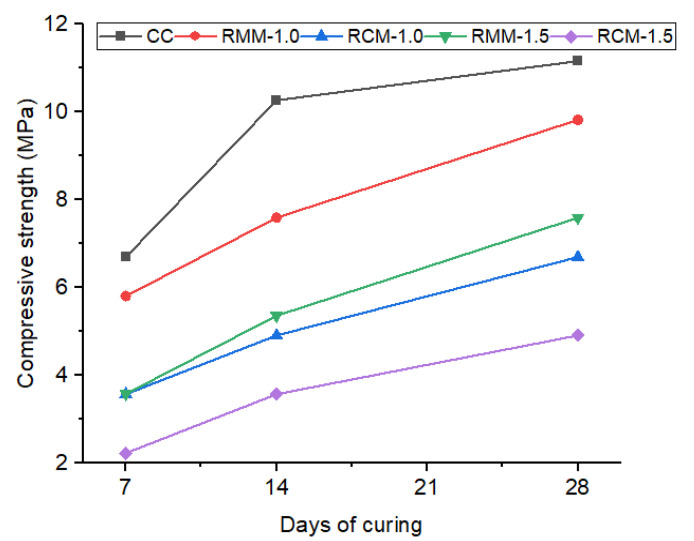
Compressive strength test results.

**Figure 6 materials-15-04520-f006:**
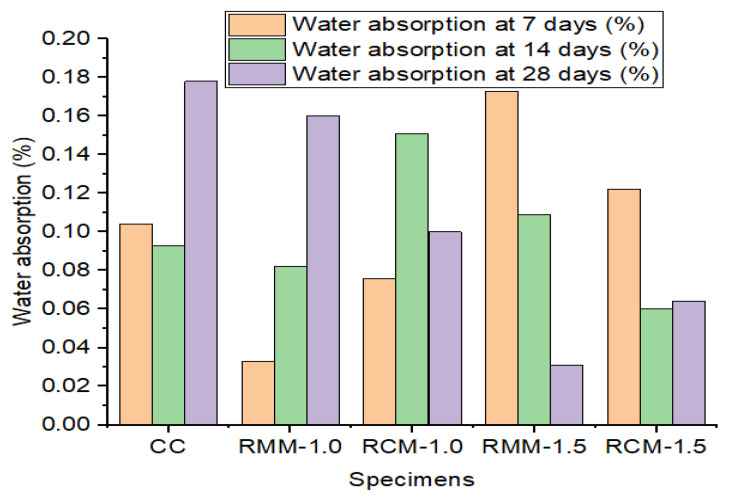
Chart showing the water absorption results of samples at 7, 14, and 28 days.

**Figure 7 materials-15-04520-f007:**
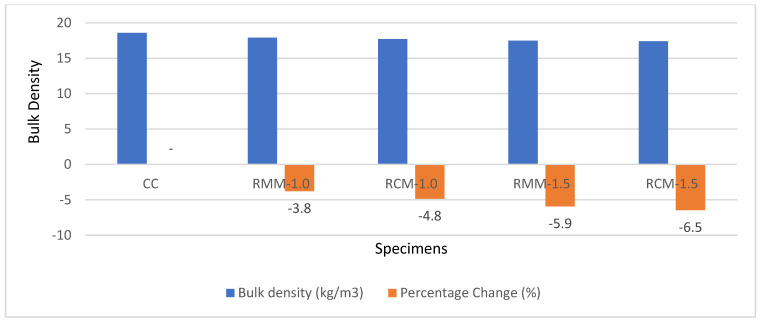
Chart showing the bulk density results.

**Figure 8 materials-15-04520-f008:**
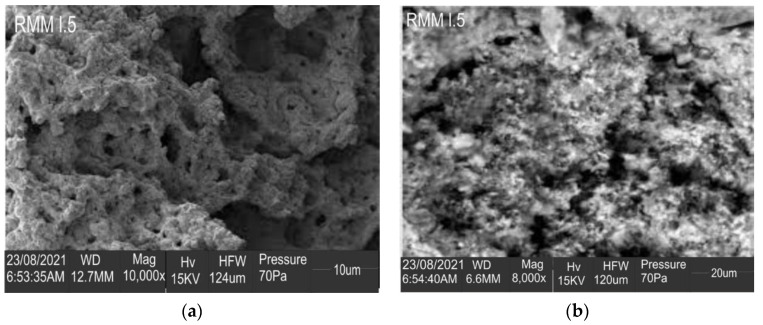
SEM of best fibre-reinforced mortar mix (**a**) at 7 days (**b**) at 28 days.

**Figure 9 materials-15-04520-f009:**
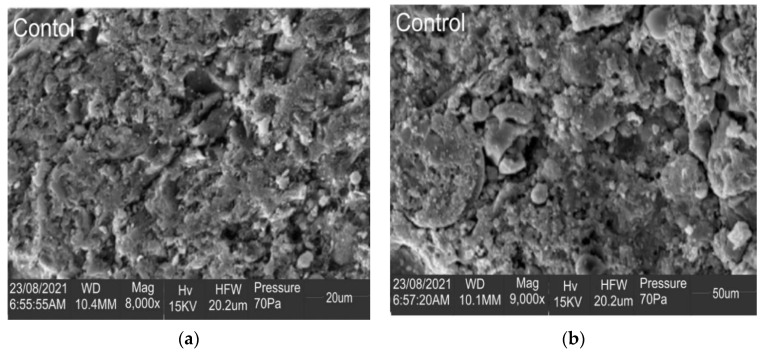
SEM of control mortar mix (**a**) at 7 days (**b**) at 28 days.

**Figure 10 materials-15-04520-f010:**
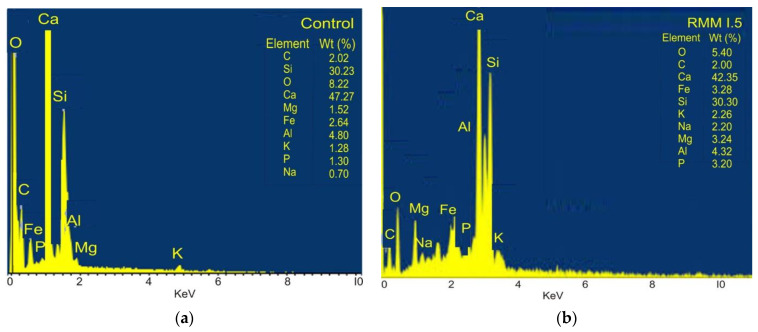
EDS analysis (**a**) control mortar mix (**b**) best fibre-reinforced mortar mix.

**Figure 11 materials-15-04520-f011:**
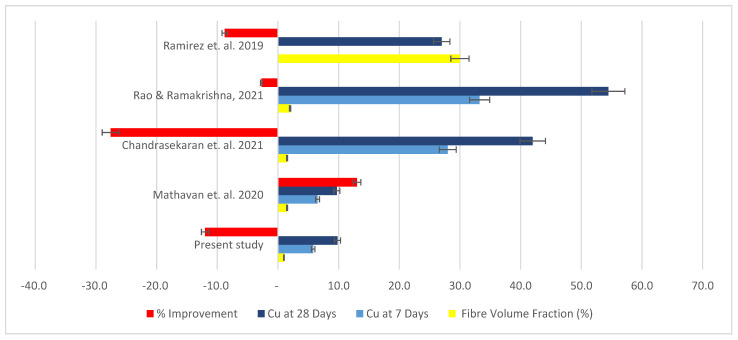
Chart expressing comparison of present study with previous studies in terms of optimal compressive strength values and improvement attained [22,32,33,34].

**Figure 12 materials-15-04520-f012:**
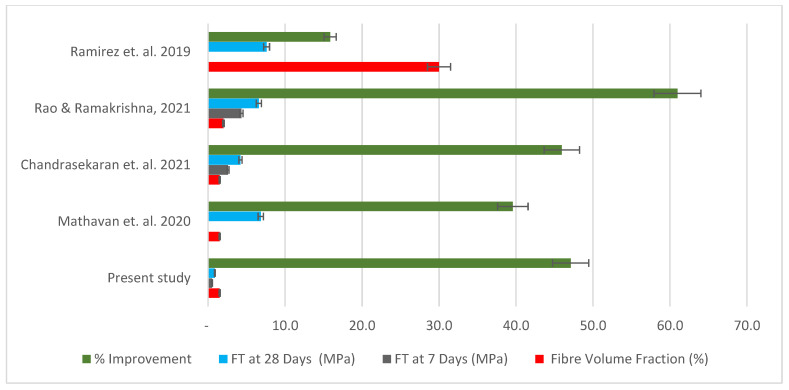
Chart expressing comparison of present study with previous studies in terms of optimal flexural strength values and improvement attained [22,32,33,34].

**Table 1 materials-15-04520-t001:** Natural fibre properties.

Fibre	Avg. Length (mm)	Density (kg/m^3^)	Tensile Strength (N/mm^2^)
Mineral wool	75	125	1932
Coconut	50	1150	145

**Table 2 materials-15-04520-t002:** Sample configurations.

Designation	Fibre Name	Adopted Mix Ratio of Cement and Fine Aggregates	Natural Fibre Content (%)
CM		1:3	-
RMM-1.0	Mineral wool		1.0
RMM-1.5	Mineral Wool		1.5
RCM-1.0	Coconut fibre		1.0
RCM-1.5	Coconut fibre		1.5

**Table 3 materials-15-04520-t003:** Slump test results.

Specimens	Content	Slump Value (mm)	Changes in Slump (%)
CM	Conventional mortar	50.0	-
RMM-1.0	1.0% mineral wool fibres	40.0	−20
RCM-1.0	1.0% coconut fibres	40.0	−20
RMM-1.5	1.5% mineral wool fibres	30.0	−40
RCM-1.5	1.5% coconut fibres	35.0	−30

**Table 4 materials-15-04520-t004:** Water absorption test results.

Specimens	Water Absorption at 7 Days (%)	Water Absorption at 14 Days (%)	Water Absorption at 28 Days (%)
CC	0.104	0.093	0.178
RMM-1.0	0.033	0.082	0.160
RCM-1.0	0.076	0.151	0.100
RMM-1.5	0.173	0.109	0.031
RCM-1.5	0.122	0.060	0.064

**Table 5 materials-15-04520-t005:** Comparison of present with previous studies in terms of optimal compressive strength values and improvement attained [22,32,33,34].

References	Specimens	Fibre Type	Fibre Volume Fraction (%)	Cu at 7 Days	Cu at 28 Days	% Improvement
Present study	RMM-1.0	Mineral wood fibre	1.0	5.81	9.82	−12.01
Mathavan et al., 2020	M_3_	Cotton	1.5	6.52	9.72	13.02
Chandrasekaran et al., 2021	C7	Coir-10 mm L	1.5	28.00	42.00	−27.59
Rao & Ramakrishna, 2021	SP5	Oil Palm (Spikelet)-15 mm L	2.0	33.24	54.47	−2.73
Ramirez et al., 2019	RW 1	Rock wool waste	30.0		27.00	−8.78

N/B: Cu represents compressive strength.

**Table 6 materials-15-04520-t006:** Comparison of present with previous studies in terms of optimal flexural strength values and improvement attained [22,32,33,34].

References	Specimens	Fibre Type	Fibre Volume Fraction (%)	F_T_ at 7 Days (MPa)	F_T_ at 28 Days (MPa)	% Improvement
Present study	RMM-1.5	Mineral wood fibre	1.5	0.53	0.88	47.10
Mathavan et al. 2020	M_9_	Polyester	1.5	-	6.84	39.58
Chandrasekaran et al. 2021	P9	Palmyra-20 mm L	1.5	2.60	4.20	45.95
Rao & Ramakrishna, 2021	SP6	Oil Palm (Spikelet)-20 mm L	2.0	4.32	6.59	60.98
Ramirez et al. 2019	RW 1	Rock wool waste	30.0		7.60	15.85

N/B: F_T_ represents flexural strength.

## Data Availability

The data presented in this study are available on request from the corresponding author.

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
