# Peer review of "Development of Fibre-Reinforced Cementitious Mortar with Mineral Wool and Coconut Fibre"

_materials, 2022, doi:10.3390/ma15134520_

Round 1

Reviewer 1 Report

The manuscript is significantly improved since the last version. Most of the issuses has been addressed. It is recommended for publication after minor revision. A few editorial changes: 

1. The format of Table 5 and Table 6 need to be improved. Table 6 is cut off. And the space between line looks not correct. 

2. All the figures need to be double check and improved. 

* Please do not use the frame for all figures.

* Figure 6, Y-axis. The number of sinificance should be consistent. So "0.2" should be "0.20", etc. 

*Figure 4, 5. In my oponion, for this type of dot-line chart, if there is no particular reason, the line between dot should be straight lines instead of curves. 

 3. Double check all the headings. They are a totoal mess.  For example:

* " 2. Materials and methods". "3. Results and Dicussion" ( in the wrong position) .  Their format should be consistent. 

* Similarly, "3.1 Test on fresh motar" "3.3.1 Slump test", "3.3.2 Compressive Strength Test" "3.3.3 Water absorption Test". Totoal mess. Please check throughout for consistency.

Author Response

Response to reviewers’ comments – Round 3

Again, the authors would like to sincerely appreciate the reviewers for their constructive critiquing of this article. The comments we received have been very helpful for improving the quality of article. In what follows, we attend to the reviewers‘ comments point by point.

Reviewer #1

The manuscript is significantly improved since the last version. Most of the issuses has been addressed. It is recommended for publication after minor revision. A few editorial changes:

  1. The format of Table 5 and Table 6 need to be improved. Table 6 is cut off. And the space between line looks not correct.

Response: Thanks to the reviewer for the positive comments. Tables 5 and 6 have been adjusted as suggested.

  1. All the figures need to be double check and improved.

* Please do not use the frame for all figures.

Response: this has been taken care of.

* Figure 6, Y-axis. The number of sinificance should be consistent. So "0.2" should be "0.20", etc.

Response: Figure 6 has been replotted

*Figure 4, 5. In my oponion, for this type of dot-line chart, if there is no particular reason, the line between dot should be straight lines instead of curves.

Response: Figures 4 and 5 have been replotted

  1. Double check all the headings. They are a totoal mess. For example:

* " 2. Materials and methods". "3. Results and Dicussion" ( in the wrong position) . Their format should be consistent.

* Similarly, "3.1 Test on fresh motar" "3.3.1 Slump test", "3.3.2 Compressive Strength Test" "3.3.3 Water absorption Test". Totoal mess. Please check throughout for consistency.

Response: All headings have been checked and formatted as suggested.

Reviewer 2 Report

Line 81 - fibers do not help repairing the microcracks but either completely stop or delay their propagation once formed.

Lines 164-166 - the dimensions for the samples given by the authors are a little bit odd considering they are referring to mortar. As per BS EN 1015-11, de mortar prisms should be 40x40x160mm. Also, figure 3 shows the mortar prisms which are smaller than 75x75x500 mm

Author Response

Response to reviewers’ comments – Round 3

Again, the authors would like to sincerely appreciate the reviewers for their constructive critiquing of this article. The comments we received have been very helpful for improving the quality of article. In what follows, we attend to the reviewers‘ comments point by point.

Reviewer #1

Line 81 - fibers do not help repairing the microcracks but either completely stop or delay their propagation once formed.

Response: Thanks to the reviewer. This section has been updated as appropriate.

Lines 164-166 - the dimensions for the samples given by the authors are a little bit odd considering they are referring to mortar. As per BS EN 1015-11, de mortar prisms should be 40x40x160mm. Also, figure 3 shows the mortar prisms which are smaller than 75x75x500 mm

Response: The dimension of the tested samples has been corrected. The samples are of 40 x 40 x 160 mm and the same are presented in Figure 3.

Reviewer 3 Report

At present, the author has well revised the manuscript, and I agree to recommend this paper for publication.

Author Response

Response to reviewers’ comments – Round 3

Again, the authors would like to sincerely appreciate the reviewers for their constructive critiquing of this article. The comments we received have been very helpful for improving the quality of article. In what follows, we attend to the reviewers‘ comments point by point.

Reviewer #2

At present, the author has well revised the manuscript, and I agree to recommend this paper for publication.

Response: The reviewer is well appreciated for a positive recommendation of this article.

Reviewer 4 Report

Line 187: "from 15 to 1.5%" should be "from 1 to 1.5%".

With respect to the flexural and compressive behaviour provide an explaination to justify the following point: Why the strength with coconut addition at 1.5% is lower than 1% and why it is the opposite in case of mineral wool fibres. It seems that the two types of fibres are characterised by a different behaviour that significantly affect the strength in a different way. Please provide an exhaustive explanation of this experimental evidence.

The other comments from the reviewer were exhaustively taken into account in the revised paper.

Author Response

Response to reviewers’ comments – Round 3

Again, the authors would like to sincerely appreciate the reviewers for their constructive critiquing of this article. The comments we received have been very helpful for improving the quality of article. In what follows, we attend to the reviewers‘ comments point by point.

Reviewer #3

Line 187: "from 15 to 1.5%" should be "from 1 to 1.5%".

Response: Thanks to the reviewer. The typo error has been corrected as appropriate.

With respect to the flexural and compressive behaviour provide an explaination to justify the following point: Why the strength with coconut addition at 1.5% is lower than 1% and why it is the opposite in case of mineral wool fibres. It seems that the two types of fibres are characterised by a different behaviour that significantly affect the strength in a different way. Please provide an exhaustive explanation of this experimental evidence.

Response: Thanks to the reviewer for the thought-provoking comment. As mentioned, strength properties reduced when coconut fibre content was increased from 1 to 1.5%. it was deduced that increasing the volume of the fibre could result in overlapping of the fibres. As such, during loading, areas where fibre laps becomes the weakest, thus creating a pathway for failure to occur in the material. The revised manuscript has been updated as appropriate.

The other comments from the reviewer were exhaustively taken into account in the revised paper.

Response: Thanks to the reviewer.

This manuscript is a resubmission of an earlier submission. The following is a list of the peer review reports and author responses from that submission.

Round 1

Reviewer 1 Report

The topic is straightforward, and if done properly, could be an interesting one. However, as it is now, the quality of the manuscript is quite low, hence does not meet the standard of this journal. 

  1. The tables and figures are in quite low quality overall. It needs significant improvement.

* Figure 3. The x-axial should be in log scale.

* Figure 4,5,6,7 need error bars to demonstrate the statistical significance.

*Figure 10 is an EDS analysis, and not an XRD. And usually an EDS figure is not that interesting. A table of element composition would suffice.

  1. The citation is a total mess. You can not switch between numbered and author-date.
  2. It is a great ideal if the waste materials such as coconut fiber and mineral wool can be utilized in mortar/concrete. However, the mortar prepared in this manuscript needs more clarification of its class and applications. The cement to sand ratio is 1:3, which is quite high, with a very high water to cement ratio of 0.55, it kept me wondering what’s the application scenario for these mortars? Although RMM-1.5 achieved 47% increase in flexural strength, it is not very helpful the compressive strength dropped 32% at the same time. Not to mention that the coconut fibers were alkali treated. From the results, it seems not worth the trouble…

4.The language can use some editing. The formatting needs to be checked and double checked throughout the manuscript.

Reviewer 2 Report

The tackles an interesting topic and aims at offering a recycling alternative to some of the by-products / wastes generated to ensure the quality of life and comfort of modern society.

The authors need to be more specific on the type of mineral wool used in the research. Mineral wool is a very generic term and can refer to both natural and synthetic origins of the material. So far, the prospective user only gets the information that it is of natural source. However, to refer to mineral wool as being natural it is a bit far fetched (as opposed for example to coconut fibers). 

It would be highly advisable to turn to some professional editing services since the English language needs extensive changes. Some of the mistakes are typing errors and should be easy to correct but the manuscript is hard to read and understand.

Line 62 - "this" instead of "these"

Line 87 - "fissures" is not a very common term used to convey the idea of micro/small cracks in mortar

Line 88 - what do you understand by "sway force"?

Line 110 - "was utilized". It is important to give the full description of the type of cement for an easier understanding of the obtained results. Moreover, the intended use of the selected mortar should also be specified.

Line 113 - "were gotten"?

Line 122 - "The determination of fresh and hardened properties were carried out in accordance to ...."

Line 133 - concrete or mortar? It is still not clear how many samples were cast for each mix in Table 1. As written, the reader understands that 15 samples were cast altogether. I think the number does not comply with the cited codes.

Figures 4 and 5 - the values of the flexural tensile and compressive strengths should be displayed on the graphs for an easier understanding. It is also suggested that the horizontal axis should represent the curing age, instead of mix designation. It would be much easier to draw comparisons between the results.

Line 134 - the first and only time in the paper sugarcane is referred to

Please give details on the storage conditions for the mortar samples until the day of testing.

Where is section 4.1? What is Section 4 named?

Section 4.2.3 - please explain a little more in detail how did you conduct water absorption tests on the mortar specimens.

Table 4 should be removed from the main body of the manuscript and submitted as supplementary material to the paper.

Please use citations as instructed in the paper template: Lines 67, 303, 304, 320 (missing all together), 333, 334.

Figures 8 and 9 - it is unclear what the authors want to transmit with those SEM images. The images seem out of focus. Lines 245-248 do not offer enough information.

The paper needs to be more organized and the authors need to pay more attention to the instructions given in the template.

Reviewer 3 Report

This paper mainly studies the feasibility of fiber reinforced mortar by adjusting the content of coconut fiber and mineral wool fiber. There are many problems in this paper. See below for more tips:

  1. The abstract is too long. The author should refine it and add some experimental results to it.
  2. Line 48-50: The author said that Indonesia, the Philippines and India were the world's largest coconut producing countries. However, the author of this paper does not have staff in the research institutions of these countries, so please explain the reasons for choosing coconut fiber as mortar reinforcement, or why not choose other materials with more abundant domestic sources for experiments?
  3. Line 56: obvious format problems.
  4. The innovation of this paper is too weak. The author needs to rewrite the introduction. At present, agricultural fibers have been widely used in the field of building materials, and have good effects on improving the mechanical properties of cement-based materials, crack control and reducing water loss. The author uses coconut fiber in this paper, which is also a kind of agricultural fiber, and there is no special innovation. Moreover, it is unreasonable for the author to say that the mixed use of mineral wool fiber and coconut fiber has a good impact on waste recycling and environmental protection.When fibers are used in cement-based materials, they are most afraid of being damaged by corrosion due to the high alkalinity inside the cement-based materials. The author should describe this problem well.
  5. Line 110: Ordinary Portland Cement, 32.5, 42.5 or 52.5?
  6. Line 117: degrees Celsius......, ℃ maybe better.
  7. Line 119-120: ......the mixture similar to fiber reinforced concrete in line with the sample configurations...... concrete? Concrete and mortar are different, the author needs to make this clear in the paper.
  8. Line 133-134: Fifteen different mortar samples were prepared which include three control concrete and twelve fibre-reinforced mortar...... The same issues as tip 6. And three control concrete should be revised to three control concretes.
  9. Section 3 Results and Discussion. This paper has only "results" and no "discussion". The author must rewrite this part.
  10. Sections 3, 4 and 5 should be merged. The current structure of the paper is difficult to read.
  11. Table 4 is too complicated. It is suggested that the author draw pictures.
  12. Line 374-375: 4. Conclusions; 6. Conclusion and Recommendation. Authors should take scientific research writing seriously.

Reviewer 4 Report

The paper, entitled “Development of Fibre-Reinforced Cementitious Mortar with Mineral Wool and Coconut Fibre” reports on the experimental characterisation of fibre reinforced mortars produced by using wool and coconut fibres. The results are interesting, and a large database is reported as comparison. However the paper need to be strongly improved in the form in order to be published. Specifically, the section of the method is completely absent, and the one concerning the materials is poor. Moreover, results are just presented without a proper discussion. Major revisions are strongly recommended in order to enrich the paper with adequate method and discussion sections.

In the Reviewer’s opinion major revisions are needed according to the comments listed as follow:

  1. The State of the Art is widely and deeply described in the Introduction section. However, several sentences are written without referring to any scientific work. Bibliography needs to be enriched with respect to this aspect.

For instance:

“Most of these synthetic fibres present ecological and medical issues during their production and usage. Furthermore, the cost of these human-made fibres (glass fibres, polypropylene fibres and steel fibres) is high.” Please cite articles reporting this information.

“At the moment, research has been led on the utilization of common plant fibres (banana, sisal, hemp and flax, jute, coconut, and oil palm) in mortar. Most of these researches have portrayed natural fibres to exhibit prospects of enhancing mechanical and strength properties of mortar. Studies have suggested that these fibres can help in the repair of fissures in cementitious composite materials; improve the post cracking behavior of mortar; expand the protection from sway force, sway ductile depletion cracking and shrinkage cracks as well as lower perviousness of mortar, thus reducing the loss of water.” Please cite articles reporting this information. Here you can find an example of study in which the beneficial effect of short curaua fibres is shown “Ferrara, G., Pepe, M., Toledo Filho, R. D., & Martinelli, E. (2021). Mechanical response and analysis of cracking process in hybrid TRM composites with flax textile and curauá fibres. Polymers, 13(5), 715.”. However, you can find plenty of similar articles in literature.

“These fibres can be treated using natural solvents, water, and dilute alkali.” Please also mention the influence of impregnation treatment on the morphology and mechanical behaviour of natural fibres that has been largely investigated in literature and proved as valuable technique to improve stiffness and bond strength.

  1. Line 110 “was be utilized” please revise the English
  2. Line 110: “Ordinary Portland Cement was be utilized…” Many studies proved that the high content of portlandite in cement-based mortar and concrete significantly reduce the durability of natural fibres inducing degradation phenomena. Please, justify the use of this type of binder. Was it done based on mechanical properties? Moreover, please specify the class of strength of the adopted cement.
  3. The proportioning of the mix design 1:3 and 0.55 is done in terms of weight? Please specify.
  4. Lines 112-118: in this paragraph the adopted fibres properties are described. However, no specific information is provided concerning their geometry. A reliable study should include at least the density, cross section area and length of the fibres. Information about mechanical properties (such as stiffness and strength) would be appreciated as well. Please, improve this section.
  5. Figure 2 seems to show the mortar specimen tested. However no specific information is given about it through the test and figure 2 is not even mentioned. Please describe in detail the number of series, the number of specimens for each series, their geometry and the type of mechanical test carried out.
  6. Section 2 is entitled as Materials and methods. Then a sub-paragraph 2.1 is added with the name of “mix preparation”. Which is the sense of adding a subparagraph if it is the only one of the section? Please add a specific sub-paragraph describing in detail the method of the experimental campaign: geometry and number of the specimens, type of test, test set-up (type of machine, loading cell, etc…), details about the test (displacement/loading control, velocity of the test). One should be able to reproduce the same experimental campaign based on the information reported in the article.
  7. Section 3.1. Sieve analysis should be reported in materials section, not in results and discussion.
  8. Line 167: Consistency… (test?)... which is… Please revise the English.
  9. Table 2. Fresh mortar with 1.0% content of fibres showed similar consistency, while those with1.5% content showed different values. Why the addition of wool fibres resulted in a mortar with lower workability with respect to coconut fibres addition? Why this phenomenon did not occur when 1.0% content of fibre is considered? Please provide an exhaustive discussion of the results explaining in detail the experimental outcomes. Please, also try to explain, for instance based on literature, if the obtained workability is acceptable considering the envisaged application of the material.
  10. Section 4.2.1 Results need to be discussed! Why coconut addition at 1.5% is lower than 1% and why it is the opposite in case of mineral wool fibres. It seems that the two types of fibres are characterised by a different behaviour that significantly affect the strength in a different way. Please provide an exhaustive explanation of this experimental evidence.
  11. Section 4.2.2 How do you explain the different strength exhibited by the two different types of fibres?
  12. Fifures 4 and 5 Please, provide the title of the y axis.
  13. Section 4.2.2 Please describe in detail how the absorption test was carried out in the Method section. Please, provide an explanation on the different behaviour of the specimens in terms of water absorption. Figure 6 shows confusing results: some specimens exhibited a water absorption rate increasing with the time, some others show the opposite. Please explain this result.
  14. Section 4.2.4. If the density is just a parameter to characterise the materials, then move this section in materials section. If not, provide a detailed discussion.
  15. Line 300 301. Please revise the English
  • Section 5. This section provides a large dataset used to compare the results of the experimental study. However, the discussion is carried out by citing some results without a clear explanation. At the end, it results really hard to understand if the experimental results are or are not in line with literature. Probably the use of charts may help for a clearer presentation of the data.
  • Conclusions: I would not list all the results for each series of specimen in this section, an overall explanation of the mechanical behaviour in compression and in bending conditions is sufficient.
  1. “more economical option”, how did you arrive to this conclusion in economic aspects were never mentioned through the text?

 As general comment, the aim of the study is the investigation of the feasibility of the utilization of natural fibres for the production of fibre-reinforced mortar. General comments concerning this aspects are needed at the end of the discussion section, based on the analysis proposed by the study.